# T2VIndexer: A Generative Video Indexer for Efficient Text-Video Retrieval

## ABSTRACT

Current text-video retrieval methods mainly rely on cross-modal matching between queries and videos to calculate their similarity scores, which are then sorted to obtain retrieval results. This method considers the matching between each candidate video and the query, but it incurs a significant time cost and will increase notably with the increase of candidates. Generative models are common in natural language processing and computer vision, and have been successfully applied in document retrieval, but their application in multimodal retrieval remains unexplored. To enhance retrieval efficiency, in this paper, we introduce a model-based video indexer named T2VIndexer, which is a sequence-to-sequence generative model directly generating video identifiers and retrieving candidate videos with constant time complexity. T2VIndexer aims to reduce retrieval time while maintaining high accuracy. To achieve this goal, we propose video identifier encoding and query-identifier augmentation approaches to represent videos as short sequences while preserving their semantic information. Our method consistently enhances the retrieval efficiency of current state-of-the-art models on four standard datasets. It enables baselines with only 30%-50% of the original retrieval time to achieve better retrieval performance on MSR-VTT (+1.0%), MSVD (+1.8%), ActivityNet (+1.5%), and DiDeMo (+0.2%). The code is available at https://anonymous.4open.science/r/T2VIndexer-40BE.

## CCS CONCEPTS

• **Information systems** → **Language models**; **Retrieval models and ranking**; **Novelty in information retrieval**.

## KEYWORDS

Deep Learning, Multi-modal Learning, Video Retrieval, Generative Model

## 1 INTRODUCTION

Given a query text description, text-video retrieval [26] aims to retrieve videos that are semantically relevant to the query. Text-video retrieval is flexible to express the user's intent and brings emerging attention for web search with the dramatic increasing of videos uploaded online every day. For a standard web search engine [20], video retrieval and ranking are two core stages. The retrieval stage first retrieves limited number of candidate videos from massive

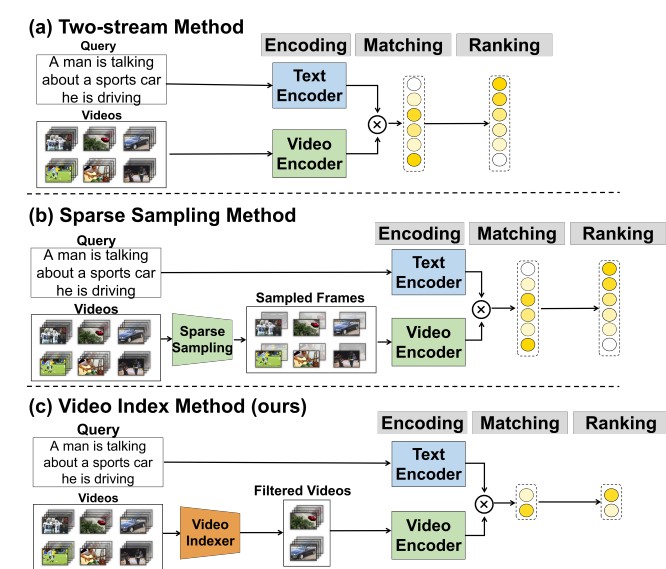

**Figure 1: (a) Two stream method with independent video and text encoders. (b) Video sparse sampling for efficiency boost. (c) Our generative video indexer for efficiency boost.**

online videos, and the following ranking stage predicts accurate ranking scores between per query and the candidate videos. Since videos have much richer and more diverse visual content compared with the query text, precise video ranking is costly for fine-grained text-video matching. Therefore, the efficiency and recall performance of video retrieval stage is essential to the fast and accurate text-video search.

Existing text-video retrieval methods can be divided into two categories, namely *one-stream* and *two-stream* approaches. One-stream approaches [32] [12] [16] adopts deep models for feature-level interactions between each text-video pair to predict its similarity score, which require online feature extraction and fail to be applied for the time-sensitive retrieval stage. Thus, the efficient two-stream approaches [9] [17] [23] are widely applied. As shown in Figure 1 (a), they encode each video and text independently into dense embeddings and then adopt simple matching functions to measure their similarity. Since there are no text-video interactions in the encoding stage, two-stream approaches allows offline data embedding extraction and alleviating online computation. Some recent works begin to focus on the issues of reducing the high computational overload of dense video embedding by sparsely sampling a few clips [12] (see Figure 1 (b)). However, all the existing solutions require to measure the query-video similarities and then rank videos for the entire video set (*i.e.*, one-to-all retrieval framework). Thus, their online retrieval time grows linearly with the increase of retrieved videos, which limits their scalability on large-scale scenarios.

To address the above issue, we explore to fundamentally change the traditional one-to-all embedding retrieval framework by a generative deep model that directly generates video identifiers and retrieves video candidates with constant time complexity. As illustrated in Figure 1 (c), our target of this work is not to propose a new model on text-video retrieval. We mainly investigate how to design a model-based indexer that effectively retrieves query-relevant video candidates, which shortens the overall retrieval time while maintaining the retrieval accuracy of state-of-the-art ranking models. To this end, we propose a sequence-to-sequence generative network that supports **T**ext query to **V**ideo candidate **Index**, named as **T2VIndexer**. The model is based on the encoder-decoder that feeds the query into the encoder and generates the identifier of the video candidate through the decoder. It is trained by query-identifier pairs that supports controllable video recall at different semantic grained. During inference, the top $K$ videos are directly retrieved by beam search and identifier constrain.

To guarantee the effectiveness of T2VIndexer, we have proposed several methods to tackle the key challenges. First, to get the semantic representations and coarse-to-fine identifiers of videos for controllable video recall, we utilize CLIP [19] to embed each video, and then cluster and encode the semantic embeddings in a hierarchical mode. Second, we propose to leverage the pre-trained multi-modal large language model [28] to generate new queries with diverse views of the video content, which augments the query-identifier pairs during training for stronger generalization ability during inference. Third, we propose to train a generative network based on T5 [4] architecture to enable the deep interactions between the query and video identifier, which enhances the cross-modal correlation learning for precise identifier prediction.

The main contributions are summarized as follows: (1) We propose a novel sequence-to-sequence generative framework as a video indexer for efficient video candidate retrieval. Our approach directly predicts the candidate videos with constant time complexity that significantly outperforms existing one-to-all embedding retrieval solutions with linear time complexity. It demonstrates the effectiveness of generative video index and sheds new light on the research on generation-based text-video retrieval mechanism. (2) We novelly propose the video identifier encoding and query-identifier augmentation approaches for learning T2VIndexer with strong generalization ability. T2VIndexer is model-agnostic and universal to cooperate with various independent-embedding approaches, which remarkably improves their retrieval efficiency with even better retrieval performance. (3) Our T2VIndexer approach is consistently effective for diverse text-video retrieval tasks. By cooperated with T2VIndex, the state-of-the-art models cost merely 30% to 50% of original retrieval time across four typical tasks. The time cost will be further reduced with the increase of retrieved videos, which impacts a broader range of text-video applications.

## 2 RELATED WORK

**Text-video retrieval.** Existing methods can be divided into two categories, called *one-stream* and *two-stream* approaches. One-stream approaches are characterized by token-level interactions based on cross-modal attention mechanisms, which are used for fine-grained video-text matching [32] [12] [16] [24]. Two-stream approaches

aim to coordinate videos and text in a unified semantic space and perform direct comparisons through distance metrics [9] [17] [23]. With the success of pre-trained image-text alignment model such as CLIP [19], this method not only surpasses interactive embedding methods in efficiency but also has significant advantages in accuracy. In addition, efficiency enhancements have focused on video sampling strategies. Some methods choose to sample the frame sparsely [12] [17]. Besides, redundancy persists within the vision tokens of each frame, diminishing the prowess of CLIP-style retrieval. CenterCLIP [31] addressed this by refining patch subdivision and selection via clustering. These innovations enhance preprocessing efficiency but do not alleviate the inherent online retrieval latency due to similarity computations and ranking cost. **Generative Model in Retrieval.** In unimodal retrieval tasks, the same efficiency issues are faced. With the success of generative models in various visual and language tasks, they have demonstrated powerful capabilities. In document retrieval, models like DSI [21] demonstrate the ability to generate identifiers using Transformer architectures, while approaches like SEAL [3] innovate by substituting string identifiers with document n-grams. The NCI [22] further refines this approach by integrating positional information into the decoding process. Image-to-image retrieval tasks have transformed these methods into visual modality. For example, IRGen [30] tokenizes images to identifiers and uses a generative model to map queries to these identifiers for direct localization, thereby improving retrieval efficiency. These methods have demonstrated powerful capabilities in unimodal retrieval. However, videos contain rich target and event. There is an obvious many-to-many problem, which means one video corresponds to multiple different descriptions from different perspectives, and a summary description corresponds to multiple different videos.

## 3 METHODOLOGY

The text-video retrieval involves a text query $t$ and a gallery of videos $V$. The objective is to retrieve videos $\{v_j\} \in V$ that are semantically relevant to the query. As shown in Figure 2, our goal is to directly retrieve targeted videos by generating the video identifiers based on natural language queries. To this end, we design a sequence-to-sequence generative model that takes query $t$ as input and outputs the video identifier for video index. We first propose a semantic-aware tree structure to encode video identifiers, called SemID, which encodes the multi-grained semantics of videos by a sequence for controllable recall while maintaining the sequence length as short as possible for fast encoding. To augment the semantic expression of queries for more generalized model learning, we propose to utilize a Multi-modal Large Language Model (MLLM) [28] to generate a set of multi-view queries for each video, thereby enriching the contextual semantics encapsulated by the SemIDs for diverse queries. The model architecture, training and inference strategies are introduced in the end.

## 3.1 Vi-SemTree for Video Identifying

The purpose of our work is to locate videos by taking query $t$ as input and outputting the most relevant video identifier. Therefore, finding a suitable identifier as the basis for video location is crucial. The identifier needs to have semantic prior information so that it

**Training Stage**

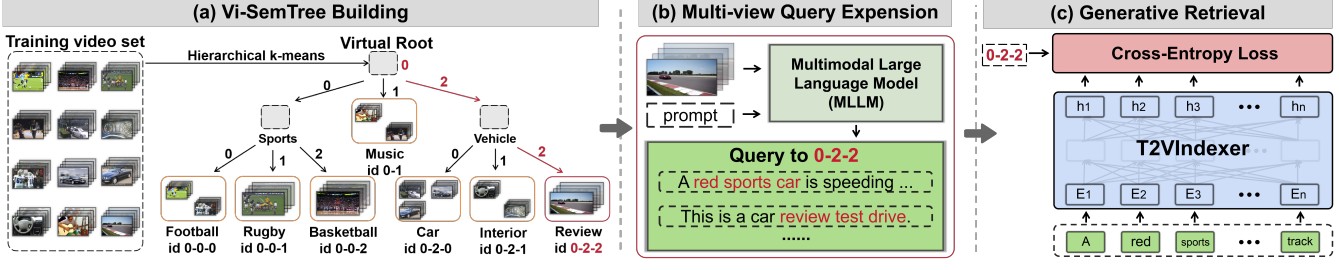

**Inference Stage**

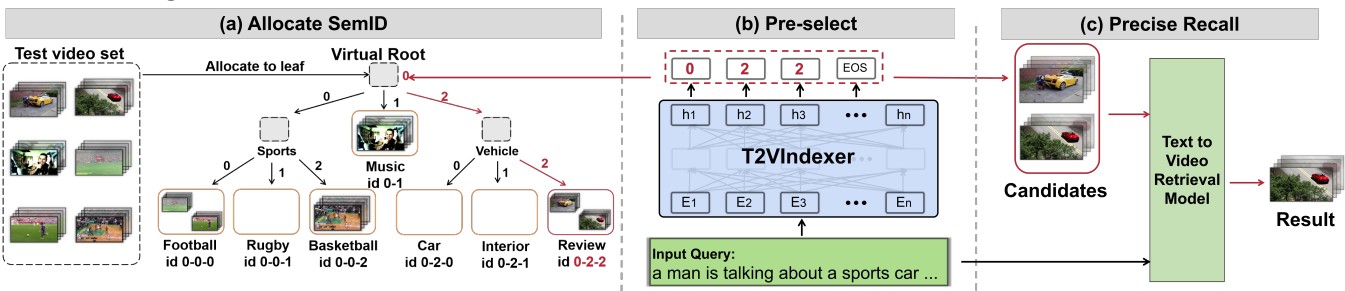

**Figure 2: An overview of our T2VIndexer. T2VIndexer uses two different strategies for the training and inference stages. For the training stage, as shown in (a), the process of dividing the training set into a tree structure is illustrated. Training stage (b) shows the process of achieving multi-view query expansion through MLLM. (c) presents the pipeline for model training. For the inference stage, the new video is first inserted into the semantic tree and assigned a SemID, and the baseline model provides the precise retrieval results.**

can reflect the content of the video, and similar semantic videos are also similar in the identifier. Moreover, the sequence length should be short enough to reduce the difficulty and complexity of model generation. Based on this consideration, we first extract the representation of each video, and construct a Video Semantic Tree (Vi-SemTree) based on the semantics of the video, and provide a SemID as an identifier for each video based on the tree structure. This approach ensures the semantic consistency of locating videos and guarantees the recall rate of the generation phase.

**Video Semantic Representation.** To construct Vi-SemTree and obtain a sequence representation of SemID, we first extract the representation of the video. Compared with pixel-level information, Vi-SemTree requires the integration of semantic-level information, which is more seamlessly integrated with the structure of natural language. To meet this requirement, we chose the image encoder of CLIP [19], which is famous for its multimodal pretraining ability, as the basic tool for our video encoding. Given a video's sequence of frames $v_f = \{f^1, f^2, ..., f^N\}$, where $N$ is the number of frames. We derive the corresponding frame representation as $\hat{f} = \{\hat{f}^1, \hat{f}^2, ..., \hat{f}^N\}$, culminating in the overall video representation $\hat{F}$, obtained through mean pooling of the individual frame representations. Conforming to the methodologies laid out by ViT [7] and CLIP [19], the output gleaned from the [class] token is utilized to represent each frame.

**Hierarchical Vi-SemTree Building.** We use a tree structure to encode videos, which helps preserve semantic information and ensures that similar semantic videos are also similar in the identifier.

Moreover, by controlling the depth $d$ of the tree, we can achieve different levels of granularity and control the sequence length. Following NCI[22], we use hierarchical $k$-means method for video feature $\hat{F}$, as shown in the training stage (a) of Figure 2. First, we use the $k$-means algorithm to divide the training set videos into $k$ clusters based on their representation similarity. Each cluster is a tree node and contains a group of semantically similar videos, which serves as a layer of the tree structure. For each cluster, if the number of videos is greater than $c$, we use the $k$-means algorithm to further divide the cluster, generating the next layer of the tree, which is more granular at the semantic level. We repeat this process until we obtain a tree structure $T$ with the root $r$, where semantically similar videos are located in the same path.

**Vi-SemTree for SemID Encoding.** To directly retrieve a video set, we introduce SemID, a unique identifier derived from the Vi-SemTree, which serves as a pivotal reference for our generative model. We define the SemID as the path $L_{\text{path}} = \{l_0, l_1, ..., l_d\}$, traversing from the root node $r$ down to a leaf node with $d$ marking tree depth. Specifically, the root node is represented as 0, serving as the start symbol. Each edge branching out from each node is numbered starting from 0, with values ranging from 0 to $k$. From top to bottom, this sequence serves as the SemID $L_{path}$.

## 3.2 Multi-view Textual Query Expansion

Due to the characteristics of the video modality, it has rich semantic information, and different descriptions can be generated from different perspectives based on different events. Therefore, a video can

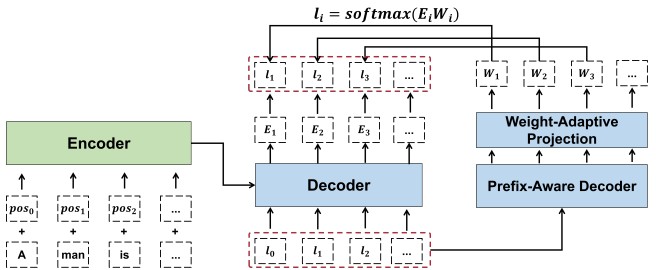

**Figure 3: Overview of the generative model of T2VIndexer.**

correspond to multiple different queries. However, the existing one video corresponds to only few descriptions, which is insufficient to cover the content of the video, making it difficult for the model to understand the semantic information of each SemID corresponding to the video group. To overcome this challenge, we designed a multi-view query expansion strategy, as shown in the Training stage (b) of Figure 2, aimed at enhancing the semantic richness.

To comprehensively capture the different elements such as targets, events, and themes that appear in the video and generate natural language descriptions that correspond to the relevant information as additional queries, Multimodal Large Language Model (MLLM) is an important tool. Unlike the single and coarse descriptions provided by dense video caption models, MLLM can generate multi-angle descriptions according to different prompts, capturing various information that appears in the video. Based on MLLM, we generated 50 different queries from different perspectives for each video, achieving coverage of semantic information.

Based on the query $t$ provided in the dataset and the extended query $\hat{t}$, the query set $Q_i = \{t_i, \hat{t}_i\}$ is associated with each video $i$ in the training set $D$. We can represent the training data pairs for the generation model as $\{(q, SemID_i), \; q \in Q_i, \; i \in D\}$, where $SemID_i$ is the $L_{path} = \{l_0, l_1, ..., l_d\}$ of the node where video $i$ is located. However, as the tree structure deepens, the semantic information of videos in different nodes becomes increasingly similar. This may lead to a decrease in the recall effect of relatively coarse-grained Queries. To merge more videos with similar semantics and simplify the generation process, we choose the truncated version of the path as SemID, represented as $L_{path}^t = \{l_0, l_1, ..., l_{d-t}\}$, to ensure efficient and semantically consistent grouping.

### 3.3 Generative Retrieval Model

To achieve direct positioning of the target video through SemID, we chose to use a sequence-to-sequence generative model to directly generate the corresponding SemID based on the input query, as shown in Figure 3. First, the input query is added with position embedding and input into the transformer encoder to obtain the representation $f_t$. The probability of generating the SemID sequence as follows to construct,

$$p(L_{path}^t | f_t) = \prod_{i=1} p(l_i | f_t, l_0, l_1, ..., l_{i-1}) \quad (1)$$

which means the next token is generated according to the sequence of previously generated tokens. This probability problem can be solved by the traditional transformer encoder-decoder structure.

The input of the encoder and the output of the decoder can be regarded as two different semantic spaces, corresponding to the natural language space and the video semantic tree space.

However, unlike standard decoding tasks, the same token appearing at different positions in the video semantic tree space has different meanings because they are in different layers of the Vi-SemTree. For example, as shown in the framework Figure 2, for SemID '$0_0 - 0_1 - 0_2$', the token '$0_1$' in the first layer represents the semantic of the category "sports", while '$0_2$' in the second layer only represents "basketball". In addition, even if they are in the same layer, the same token '$0_2$' in SemID '$0_0 - 0_1 - 0_2$' and '$0_0 - 2_1 - 0_2$' expresses different concept due to their different prefixes. In order to identify different representations at different positions during decoding, we were inspired by NCI [22] in document retrieval and used the Prefix-Aware Weight-Adaptor (PAWA) decoder, as shown in Figure 3.

Unlike the standard transformer decoder, the PAWA decoder uses different parameters when generating tokens at different positions, and the parameters between different steps are not shared. The specific settings are as follows. First, the encoder encodes the query to obtain the encoding $x$, and decoder output $E$ as follows,

$$E_i = \text{Decoder}(x, l_0, l_1, ..., l_{i-1}; \theta_i) \quad (2)$$

where $\theta_i$ represents the parameters of each decoding step, and the parameters of each step are different, distinguishing the semantic of different position tokens. In addition, to further enhance the prefix information as the basis for generation, the PAWA decoder further modifies the linear classification layers based on the prefix sequence. Specifically, instead of using the same projection weight $W$ in the linear classification layer, the PAWA decoder uses an additional decoder to generate different weights for each position,

$$E_i' = \text{Decoder'}(l_0, l_1, ..., l_{i-1}; \theta_i') \quad (3)$$

$$W_i = \text{Linear}(E_i') \quad (4)$$

where $\theta_i'$ represents the parameters of the decoder that generates the weights, and $W_i$ is the generated weight matrix for the corresponding classifier. Finally, the i-th token is represented as $l_i$, calculated by softmax($E_i W_i$).

### 3.4 Model Training and Inference

**Training Loss.** The loss function for a set of training examples $\mathcal{D} = \{(q, L_{path})\}$, consisting of queries (training queries and expansion queries) and video SemID, can be expressed as follows,

$$\mathcal{L}(\theta) = \sum_{(q, L_{path})}^{\mathcal{D}} \log(L_{path} | q, \theta) \quad (5)$$

where $\log(L_{path} | q, \theta)$ represents the probability of generating SemID based on $q$, and is a standard sequence-to-sequence cross-entropy loss with teacher forcing.

**Coarse-to-fine Inference.** During the training stage, since the videos in the test set are not visible, this part of the videos is not assigned a SemID. In order to cover test set, a new video needs to be assigned a SemID as its identifier as shown in Figure 2 Inference Stage. The new video will get the representation in the same way as the training set using CLIP encoder, and then the video similarity

with the leaf nodes in the tree will be calculated. The video will be inserted into the leaf node with the highest similarity and inherit the SemID of the leaf node. For the input query $q$, the probability $p(L^t_{path}|q, \theta)$ is calculated by the generative model trained, and the target SemID is obtained to achieve direct positioning of a group of target videos. In the decoding process, considering the semantic summarization ability of natural language query, which makes videos that meet the description may not be in the same path, we use the Beam Search algorithm for decoding. This enables us to retrieve the top k SemIDs that meet the description, to adapt to this one-to-many issue. Based on the generated SemIDs, we obtain a small candidate set $V_{cand} = \{V_{SemID_1}, V_{SemID_2}, ..., V_{SemID_k}\}$ containing the target videos, where $SemID_i$ represents a group of videos corresponding to the i-th top SemID generated.

Due to the limitations of the first stage, it is currently not feasible to achieve precise retrieval based on the query. In order to meet the requirements of the task and achieve precise retrieval, fine-tuning of $V_{cand}$ is required. For this purpose, we introduce a two-stage retrieval architecture and use the existing text-video retrieval model for precise retrieval in $V_{cand}$. T2VIndexer can be integrated with the existing retrieval model without adjusting the training parameters. For a text-video retrieval model $M_{base}$, the similarity $s$ between $q$ and each video $v$ in $V_{cand}$ is calculated, and the final retrieval result is obtained by ranking $s$. Overall, the entire retrieval process can be divided into two main stages: Pre-select based on generative models and Precise Recall based on contrastive learning models.

## 4 EXPERIMENTS

**Datasets and Evaluation Metrics.** We validate our model on four dataset: MSR-VTT, MSVD, DiDeMo, and ActivityNet Caption. MSR-VTT [26] encompasses 10,000 videos, paired with 200,000 captions. We employ the Training-9k variant, following the data splits proposed by [10]. MSVD [5] contains 1,970 videos, and a wealth of approximately 40 associated English sentences per video. Train, validation and test splits contain 1,200, 100, and 670 videos, respectively. DiDeMo [1] contains 10,000 videos annotated with 40,000 sentences. We evaluate video-paragraph retrieval following [15], [13] and [2], where all sentence descriptions for a video are concatenated into a single query. ActivityNet [11] consists of 20,000 YouTube video. We follow the setting from [29] to concatenate all the descriptions of a video to form a paragraph and evaluate the model with video-paragraph retrieval.

We adopt standard retrieval metrics, namely recall at rank K (R@K), calculates the percentage of instances where the correct result is successfully retrieved within top K.

**Implementation Details.** We utilized the image encoder from the pre-trained CLIP (Vit B/32) model. For constructing the Vi-SemTree, we opted for $k$-means algorithm [18], setting both k and c to 30. SemID truncation length $t$ set to 2 and select top 11 beam search results from generative model. The encoder parameters were initialized using the T5 pre-trained model [4], while the decoder parameters were randomly initialized. During training, the learning rate was set to $2 \times 10^{-4}$ for the encoder and $1 \times 10^{-4}$ for the decoder. We utilized 8 NVIDIA V100-32GB GPUs, with a batch size of 16 per GPU and a dropout ratio of 0.1.

### 4.1 Efficiency of T2VIndexer

The purpose of T2VIndexer is to improve retrieval efficiency while maintaining accuracy. In Table 1, we analyzed the accuracy and efficiency of T2VIndexer under different candidate sets with a single RTX3090 GPU and 8255C CPU. In the efficiency analysis, we imposed some restrictions to simulate real application scenarios. First, the time cost from receiving the query was calculated, without considering the offline phase, such as the construction of Vi-SemTree and the allocation of SemID. Second, each query in the Test set was retrieved one by one to return the target video, instead of obtaining all Queries and returning the overall results at once. From the Table 1, it can be seen that T2VIndexer significantly reduces inference time while maintaining the baseline effect. For example, compared with the traditional method under 1000 candidate videos, T2VIndexer reduces the time cost by 50%. The efficiency improvement increases gradually with the size of the candidate set. Under 10,000 candidate videos, the time compression reaches 30%. In addition, we also analyzed the performance of three other datasets, as shown in Table 2.

Further analysis shows the efficiency improvement is due to the structure of Vi-SemTree. When the candidate set expands 10,000, the traditional method needs to process an additional 9000 data, perform a large number of similarity calculations and sorting. For T2VIndexer, the added 9000 videos will be distributed to each leaf node, and cost of generative model generating SemID remains unchanged, which ultimately improves efficiency. For example, if Vi-SemTree has 100 leaf nodes, for 10,000 candidate videos, there are an average of 100 target videos per leaf node, which significantly reduces the retrieval pressure of the baseline. This means that the T2VIndexer does not suffer from the same scalability issues as traditional methods, and allows for a more distribution of data.

### 4.2 Evaluating on Large-Scale Dataset

To further investigate the effectiveness of T2VIndexer in real retrieval scenarios, we evaluate on larger scale data. Due to the limited size of the existing dataset test sets, which mostly consist of 1000 candidate videos, we decided to redivide the TGIF dataset [14] into 50,000 training data and 50,000 testing data in a 5:5 ratio. Both the baseline and the T2VIndexer generative model will be trained solely on the 50,000 training data. Table 3 displays our test results, with each block representing a set of test results. It is evident from the results that in large-scale retrieval scenarios, T2VIndexer demonstrates more significant improvements in both efficiency and accuracy compared to smaller-scale data.

The improvements in both efficiency and accuracy primarily stem from the pre-select mechanism employed by T2VIndexer. T2VIndexer operates with constant-time complexity to generate an ID sequence, corresponding to a subset of videos on a small scale. And the base model conducts similarity calculations and sorting solely within this small-scale video subset, T2VIndexer significantly reduces computational overhead by avoiding individual matching, thus enhancing retrieval efficiency. Furthermore, the pre-select mechanism aids in pinpointing the target video, eliminating a majority of irrelevant videos in advance, thereby reducing noise-induced interference on the baseline and effectively acting as a filter to enhance retrieval accuracy.

**Table 1: Evaluation of the inference costs on MSR-VTT dataset. We report the R@1 metric for the text-to-video task. Blue means stage 1 generative time cost, and red means stage 2 time cost. The improvement of inference time refers to the compression ratio of T2VIndexer, which is obtained by dividing the inference time with T2VIndexer by the inference time of the baseline.**

| Method | #Candidate 1000 | | #Candidate 3000 | | #Candidate 5000 | | #Candidate 10000 | |
|---|---|---|---|---|---|---|---|---|
| | Inference Time(ms)↓ | R@1↑ | Inference Time(ms)↓ | R@1↑ | Inference Time(ms)↓ | R@1↑ | Inference Time(ms)↓ | R@1↑ |
| CLIP4Clip [17] | 220 | 44.7 | 681 | 36.2 | 1139 | 31.6 | 2341 | 25.4 |
| CLIP4Clip+T2VIndexer | 105 (52+53) | 47.8 | 218 (52+166) | 40.3 | 336 (52+284) | 35.1 | 639 (52+587) | 27.3 |
| Improvement | 0.47 | 3.1 | 0.32 | 4.1 | 0.29 | 3.5 | 0.27 | 1.9 |
| mPLUG [25] | 189 | 53.0 | 562 | 44.7 | 961 | 38.3 | 1741 | 29.1 |
| mPLUG+T2VIndexer | 97 (52+45) | 54.3 | 189 (52+137) | 45.8 | 290 (52+238) | 40.4 | 485 (52+433) | 30.1 |
| Improvement | 0.51 | 1.3 | 0.34 | 1.1 | 0.30 | 2.1 | 0.28 | 1.0 |
| CLIP-VIP [27] | 192 | 54.1 | 579 | 48.4 | 968 | 40.1 | 1941 | 31.5 |
| CLIP-VIP+T2VIndexer | 98 (52+46) | 55.1 | 194 (52+142) | 49.3 | 295 (52+243) | 41.3 | 537 (52+485) | 33.1 |
| Improvement | 0.51 | 1.0 | 0.34 | 0.9 | 0.30 | 1.2 | 0.28 | 1.6 |

**Table 2: Evaluation of the inference costs on MSVD, DiDeMo and ActivityNet dataset. We report the R@1 metric for the text-to-video task. Blue means stage 1 generative time cost, and red means stage 2 time cost.**

| Method | MSVD | | DiDeMo | | ActivityNet | |
|---|---|---|---|---|---|---|
| | Inference Time(ms)↓ | R@1↑ | Inference Time(ms)↓ | R@1↑ | Inference Time(ms)↓ | R@1↑ |
| CLIP4Clip [17] | 209 | 45.1 | 225 | 43.4 | 1041 | 40.2 |
| T2VIndexer+CLIP4Clip | 106 (52+54) | 47.4 | 102 (52+50) | 46.3 | 311 (52+259) | 42.6 |
| Improvement | 0.51 | 2.3 | 0.45 | 2.9 | 0.30 | 2.4 |
| mPLUG [25] | 193 | 53.6 | 201 | 56.4 | 923 | 52.2 |
| mPLUG+T2VIndexer | 102 (52+50) | 55.4 | 98 (52+46) | 56.6 | 292 (52+241) | 53.5 |
| Improvement | 0.53 | 1.8 | 0.48 | 0.2 | 0.32 | 1.3 |
| CLIP-VIP [27] | 191 | 52.3 | 199 | 50.5 | 934 | 53.4 |
| CLIP-VIP+T2VIndexer | 101 (52+49) | 54.0 | 97 (52+45) | 51.9 | 299 (52+247) | 54.9 |
| Improvement | 0.53 | 1.7 | 0.49 | 1.4 | 0.32 | 1.5 |

**Table 3: Evaluation of the inference costs on large-scale dataset split from TGIF [14]. We report the R@1 metric for the text-to-video task. Blue means stage 1 generative time cost, and red means stage 2 time cost.**

| Method | Inference Time(ms)↓ | R@1↑ |
|---|---|---|
| Clip4Clip [17] | 12322 | 13.4 |
| Clip4Clip+T2VIndexer | 3064(52+3012) | 16.5 |
| Improvement | 0.25 | 3.1 |
| mPLUG [25] | 10249 | 18.2 |
| mPLUG+T2VIndexer | 2831(52+2779) | 21.1 |
| Improvement | 0.27 | 2.9 |
| CLIP-VIP [27] | 10414 | 19.7 |
| CLIP-VIP+T2VIndexer | 2902(52+2850) | 22.4 |
| Improvement | 0.28 | 2.7 |

## 4.3 State-of-the-Art Comparison

Table 4 is segmented into three blocks, each representing a distinct category of methods: Interactive embedding methods, Independent embedding method, and ours T2VIndexer, which is implemented based on different baselines. Observing the first two blocks, two-stream approaches usually exhibit excellent performance, which is mainly attributed to the powerful pretraining ability of CLIP [19]. In the third block, we constructed T2VIndexer based on CLIP4CLIP [17], mPLUG [25], and CLIP-VIP [27]. The original performance of these three models improved in turn, providing a basis for testing the effectiveness of T2VIndexer on different baselines with different degrees of effectiveness. As shown in Table 4,

T2VIndexer achieved significant improvement with the relatively low-performance CLIP4CLIP model, increasing R@1 by 3.3% on the MSR-VTT dataset. However, as the baseline model improved, the accuracy gain of T2VIndexer decreased, and the maximum improvement of the CLIP-VIP model was 1.0%.

This can be attributed to the candidates provided by T2VIndexer to baseline models. T2VIndexer provides the same set of candidates to the baseline models for a given query, effectively eliminating a considerable number of irrelevant videos. For lower-performing models, this removal of irrelevant videos is useful, significantly reducing the input noise and enhancing accuracy. However, higher-performing models are less sensitive to noise and can distinguish relevant content more effectively, resulting in relatively lower improvements when assisted by T2VIndexer.

## 4.4 Ablation Study on Model Structure

To further investigate the impact of different components on the model's performance, we report the ablation results on the MSR-VTT dataset in Table 5. (1) **Without query expansion (w/o query expansion)** This part has the most significant impact on the model's results. During the SemID generation process by T2VIndexer, the original video is not seen. If the semantics of the original video are not injected into SemID through queries during the training phase, the model will fail to establish a relationship between the text and SemID and will not be able to correctly generate SemID for queries not seen during training. (2) **Without Multi-view query expansion (w/o Multi-view query expansion)** This indicates not using an MLLM (Multilingual Language Models) to generate multi-view

**Table 4: Comparison with Existing One-stream approaches and Two-stream approaches. Our Re-implemented methods are denoted by the superscript '*'. The highest retrieval recall in each block is marked with underline. The recall of our models is marked with blue color when it is better than the baseline model.**

| Methods | MSR-VTT 1k | | | | MSVD | | | | DiDeMo | | | | ActivityNet Caption | | | |
|---|---|---|---|---|---|---|---|---|---|---|---|---|---|---|---|---|
| | R@1 | R@5 | R@10 | R@sum | R@1 | R@5 | R@10 | R@sum | R@1 | R@5 | R@10 | R@sum | R@1 | R@5 | R@10 | R@sum |
| *One-stream approaches* | | | | | | | | | | | | | | | | |
| UniVL [16] | 21.2 | 49.6 | 63.1 | 133.9 | - | - | - | - | - | - | - | - | - | - | - | - |
| ClipBERT [12] | 22.0 | 46.8 | 59.9 | 128.7 | - | - | - | - | 20.4 | 48.0 | 60.8 | 129.2 | 21.3 | 49.0 | 63.5 | 133.8 |
| VLM [24] | 28.1 | 55.5 | 67.4 | 151.0 | - | - | - | - | - | - | - | - | - | - | - | - |
| *Two-stream approaches* | | | | | | | | | | | | | | | | |
| MMT [9] | 26.6 | 57.1 | 69.6 | 153.3 | - | - | - | - | - | - | - | - | - | 61.4 | - | - |
| Frozen [2] | 31.0 | 59.5 | 70.5 | 161.0 | 33.7 | 64.7 | 76.3 | 174.7 | 34.6 | 65.0 | 74.7 | 174.3 | 28.8 | 60.9 | - | - |
| CLIP4Clip [17] | 44.5 | 71.4 | 81.6 | 197.5 | 45.2 | 75.5 | 84.3 | 205.0 | 43.4 | 70.2 | 80.6 | 194.2 | 40.5 | 72.4 | - | - |
| CAMoE [6] | 44.6 | 72.6 | 81.8 | 199.0 | 46.9 | 76.1 | 85.5 | 208.5 | - | - | - | - | - | - | - | - |
| CLIP2Video [8] | 45.6 | 72.6 | 81.7 | 199.9 | 47.0 | 76.8 | 85.9 | 209.7 | - | - | - | - | - | - | - | - |
| Cap4Video [23] | 51.4 | 75.7 | 83.9 | 211.0 | 51.8 | 80.8 | 88.3 | 220.9 | 52.0 | 79.4 | 87.5 | 218.9 | - | - | - | - |
| mPLUG [25] | 53.1 | 77.6 | 84.7 | 215.4 | - | - | - | - | 56.4 | 79.1 | 85.2 | 220.7 | - | - | - | - |
| CLIP-VIP [27] | 54.2 | 77.2 | 84.8 | 216.2 | - | - | - | - | 50.5 | 78.4 | 87.1 | 216.0 | 53.4 | 81.4 | 90.0 | 224.8 |
| *Ours* | | | | | | | | | | | | | | | | |
| CLIP4Clip* | 44.5 | 71.0 | 81.6 | 197.1 | 45.1 | 75.6 | 83.9 | 204.6 | 43.4 | 70.1 | 80.1 | 193.6 | 40.2 | 72.4 | 80.4 | 193.0 |
| T2VIndexer+CLIP4Clip* | 47.8 | 72.2 | 82.4 | 202.4 | 47.4 | 76.4 | 85.1 | 208.9 | 46.3 | 72.4 | 83.1 | 199.8 | 42.6 | 73.5 | 80.9 | 197.0 |
| Improvement | +3.3 | +2.2 | +0.8 | +5.3 | +2.3 | +0.8 | +1.2 | +4.3 | +2.9 | +2.3 | +1.0 | +6.2 | +2.4 | +1.1 | +0.5 | +4.0 |
| mPLUG* | 53.0 | 77.4 | 82.3 | 212.7 | 53.6 | 81.4 | 88.3 | 223.3 | 56.4 | 79.0 | 84.3 | 219.7 | 52.2 | 80.8 | 89.3 | 222.3 |
| T2VIndexer+mPLUG* | 54.3 | 77.7 | 82.5 | 214.5 | 55.4 | 81.9 | 88.5 | 225.8 | 56.6 | 79.1 | 84.1 | 219.8 | 53.5 | 81.1 | 89.5 | 224.1 |
| Improvement | +1.3 | +0.3 | +0.2 | +1.8 | +1.8 | +0.5 | +0.2 | +2.5 | +0.2 | +0.1 | -0.2 | +0.1 | +1.3 | +0.3 | +0.2 | +1.8 |
| CLIP-VIP* | 54.1 | 77.0 | 84.7 | 215.8 | 52.3 | 81.6 | 88.2 | 222.1 | 50.5 | 78.3 | 86.6 | 215.4 | 53.4 | 82.3 | 89.7 | 225.4 |
| T2VIndexer+CLIP-VIP* | 55.1 | 77.2 | 85.0 | 217.3 | 54.0 | 81.3 | 88.3 | 223.6 | 51.9 | 79.2 | 87.1 | 218.2 | 54.9 | 82.5 | 90.0 | 227.4 |
| Improvement | +1.0 | +0.2 | +0.3 | +1.5 | +1.7 | -0.3 | +0.1 | +1.5 | +1.4 | +0.9 | +0.5 | +2.8 | +1.5 | +0.2 | +0.3 | +2.0 |

**Table 5: Ablation Study on MSR-VTT-1kA**

| Method | R@1 | R@5 | R@10 |
|---|---|---|---|
| **T2VIndexer+CLIP-VIP (full model)** | **55.1** | **77.2** | **85.0** |
| w/o Query expansion | 45.2 | 69.3 | 77.9 |
| w/o Multi-view query expansion | 52.8 | 75.3 | 83.1 |
| w/o Vi-SemTree and SemID | 54.0 | 76.9 | 84.6 |

**Table 6: Ablation Study on MLLMs**

| MLLM | R@1 | R@5 | R@10 |
|---|---|---|---|
| mPLUG-owl | 55.1 | 77.2 | 85.0 |
| Minigpt-4 | 55.1 | 77.0 | 85.1 |
| LLaVA | 55.3 | 77.5 | 85.3 |

descriptions, and only using a dense caption model for generating descriptions. The results suggest that the semantic expansion of SemID allows the model to learn richer information, which can better apply SemID to the test set. (3) **Without Vi-SemTree and SemID (w/o Vi-SemTree and SemID)** Moreover, the experiment without Vi-SemTree and SemID organization confirms our theoretical premise that structured pre-injection of prior knowledge facilitates superior generalization.

## 4.5 Ablation Study on Different MLLMs

Utilizing Multi-Modal Large Language Models (MLLMs) for query expansion effectively enhances the model's generalization capabilities. Various MLLMs show little difference in the quality of query expansion, so the model's effectiveness does not rely on a specific MLLM. Apart from mPLUG-owl tested in the paper, we have also conducted supplementary tests with Minigpt-4 and LLaVA. As evident from the results in Table 6 on the MSR-VTT dataset, different MLLMs have a minor impact on retrieval accuracy.

**Table 7: Ablation Study on different video feature extractors on MSR-VTT with CLIP-VIP as baseline.**

| Feature Extractor | R@1 | R@5 | R@10 |
|---|---|---|---|
| S3D | 51.3 | 72.4 | 81.6 |
| CLIP image encoder | 55.1 | 77.2 | 85.0 |
| VideoMAE | 55.4 | 77.8 | 85.6 |

## 4.6 Ablation Study on Video Feature Extractor

Different methods of video feature extraction will affect the structure of the Vi-Sem tree, thereby influencing the overall model's effectiveness. In Table7, we tested three different video feature extraction methods: S3D based on pixel features, CLIP image encoder based on semantic features of frames, and VideoMAE pre-trained for video recognition. From the results comparison, it can be observed that the method based on pixel features performs the worst, even falling below the baseline model CLIP-VIP. This is mainly because in this case, the tree structure reflects pixel information rather than the more relevant semantic information associated with natural language, resulting in poorer performance in the video pre-select stage, failing to return clusters containing the correct videos to the precise recall stage.

## 4.7 Generative Result Visualization

We further demonstrated the ability of T2VIndexer to locate target videos through visualization. Three examples are shown in Figure 4, where the SemID generated by T2VIndexer has a high semantic similarity with the target SemID, even in wrong mapping cases. Specifically, for the query "video game clip showing here different characters", the SemID generated by T2VIndexer, 0-9-21, has a stronger matching relationship with the query than ground-truth

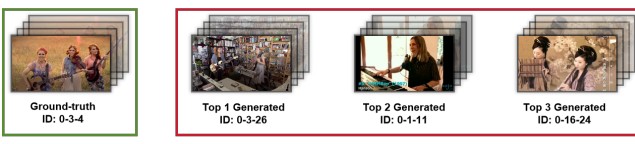

Query: woman playing instruments in a field for a music video

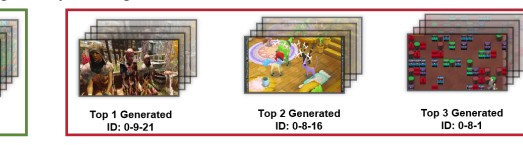

Query: video game clip showing here different charcters

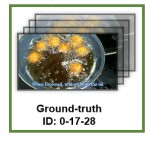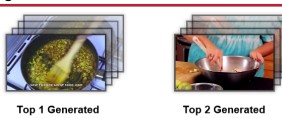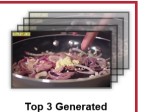

Query: someone is frying food

**Figure 4: Visualization of the difference between Generative Results and Ground-truth. We show the top-3 Generated SemIDs for each text query. The truly matched results are marked in green boxes and the falsely matched results are in red boxes.**

video. This indicates that the model has effectively learned the mapping between natural language space and SemID space, achieving retrieve target videos directly.

## 4.8 Parameter Analysis

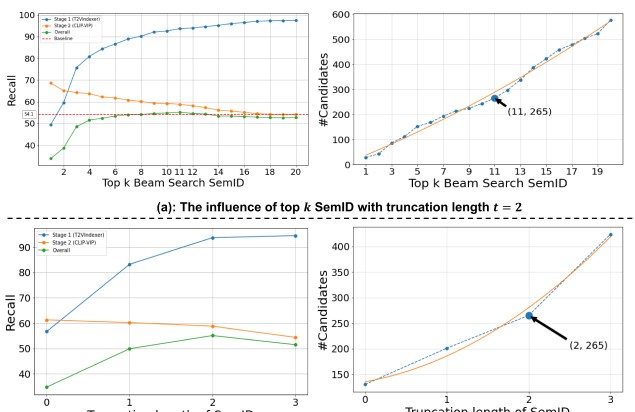

**Figure 5: Different truncation length and top k SemID for T2VIndexer on MSR-VTT-1kA.**

We studied the impact of different model configurations. Figure 5(a) shows the effect of the top $k$ SemIDs with the highest probability by beam search at $t = 2$ on recall and the number of candidates. As $k$ increases, the Recall of the second stage gradually decreases due to the increase in the number of candidate sets in the second stage. The overall Recall shows an upward trend followed by a downward trend, with the optimal recall of 55.1 achieved at

**Table 8: Ensemble framework comparative analysis. Pre-select size represents the number of videos pre-selected in the first stage.**

| Ensemble framework | Pre-select size 50 | | | Pre-select size 265 | | |
|---|---|---|---|---|---|---|
| | Stage 1 Recall | Overall R@1 | Inference Time (ms) | Stage 1 Recall | Overall R@1 | Inference Time (ms) |
| mPLUG+CLIP-VIP | **93.4** | **55.6** | 198(189+9) | **97.2** | **57.8** | 235(189+46) |
| T2VIndexer+CLIP-VIP | 42.9 | 21.4 | **61(52+9)** | 93.7 | 55.1 | **98(52+46)** |

$k = 11$. Figure 5(b) analyzes the effect of different $t$ on Recall and the number of candidate sets when $k = 11$. It also shows an upward trend with the optimal effect achieved at $t = 2$.

## 5 LIMITATION AND FUTURE WORK

Although T2VIndexer has achieved certain results in efficient text-video retrieval, there still exists performance limitations. T2VIndexer consists of two stages: pre-select and precise retrieval, which involve different models. This approach is similar to the ensemble architecture. To further analyze the effectiveness of T2VIndexer, we built a two-stage ensemble architecture based on the existing sota models mPLUG and CLIP-VIP for comparative analysis, as shown in Table 8. It can be seen that for existing ranking-based models, the ensemble form can improve the accuracy of retrieval at the expense of efficiency, which is superior to the generative method of T2VIndexer in terms of effect. However, existing models need to process all candidate sets when performing Pre-select, while T2VIndexer can directly locate the candidate set, which has a significant advantage in efficiency. In addition, the existing pipline cannot efficiently retrieve new videos that have not been seen when constructing Vi-SemTree. For each new video, it is necessary to insert it into a leaf node to obtain the corresponding SemID. This operation involves a large number of similarity calculations and sorting, which has a significant time cost. For example, inserting videos in the MSR-VTT test set into the tree takes an average of 200 ms per video to assign SemID, which reduce the flexibility.

To further improve the reliability, our future work will focus on improving the accuracy of the generative stage to achieve more precise localization. At the same time, we aim to reduce the time cost of new data and improve flexibility.

## 6 CONCLUSION

In this paper, we propose T2VIndexer, a model-based video indexer that generates video identifiers directly and retrieves candidate videos with constant time complexity, in order to shorten the overall retrieval time while maintaining the retrieval accuracy of the base model. We use hierarchical clustering to organize videos into a tree structure called Vi-SemTree, which contains multiple layers corresponding to relationships from coarse to fine. We specifically trained a generative model for Vi-SemTree paths, correctly mapping natural language space and video semantic space. T2VIndexer is model-independent and can be seamlessly integrated with existing methods. However, the retrieval effect of the generative model is currently limited, and our future work will focus on improving the accuracy of the generative retrieval to achieve precise retrieval, and further improve the flexibility of the model when receiving new videos and reduce preprocessing time.

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
