# OpenReview forum: "T2VIndexer: A Generative Video Indexer for Efficient Text-Video Retrieval"
_acmmm.org/ACMMM/2024/Conference — MM2024 Oral_

### Official Review · Reviewer_AR4B · 2024-05-25

**Rating:** 5
**Confidence:** 3

**Summary:**

In this paper, the authors highlight a problem inherent to current text-video retrieval solutions, that is the cost of retrieval and ranking. Current solutions (be it single-stream or two-stream) follow a one-against-all framework, meaning they encode the query and compare it to the encoding of each video in the gallery, and then rank according to a similarity metric. The authors propose a pipeline composed of three main steps. First, they extract CLIP-based features of the videos in the training set, and create a hierarchical representation using k-means. The resulting structure is a tree, where leaf nodes represent specific topics (e.g. basketball or volleyball) and parent nodes represent higher level topics (e.g. sports), and so on. The path to the leaf (called SemID) is used as an encoding of the group of videos with that topic. Second, they expand the training set with artificially generated captions obtained through a multimodal LLM. Finally, a T5 is used to generate the path along the tree given the input query. At inference time, the process starts with assigning the SemID to the video based on its CLIP-based representation. Then, the query given to the trained generative model which predicts top-k SemIDs through beam search. The videos in that leaf are then collected and sorted based on existing text-to-video retrieval models. The experimental setup involves four well-known datasets and three recent text-video retrieval methods. The results under different amount of videos in the test gallery confirm that the proposed approach leads to great time reduction (with favorable scaling) while maintaining high performance. State-of-the-art comparison also shows their approach leads to improvements in the three selected methods even in the full-test scenario. Ablation and qualitative studies also show robustness in their design choices. The authors then conclude highlighting a limitation of their work.

**Strengths:**

Strengths:

- The problem is relatively under-explored, since recent works on video retrieval mostly focus on improving performance, without analyzing the effect on retrieval efficiency. Moreover, the proposed approach is very interesting and aligned with recent trends in content generation, although used in a somehow unexpected way.

- The text is clear enough and easy to follow, although there are some parts which are not clearly explained (more in the limitation/weaknesses). The idea is very interesting and I believe can be applied to other multimedia domains, meaning that it can have impact on the community as a whole.

- The results are not only interesting but also extensive, in-depth, and well-presented.

**Limitations:**

Weaknesses:

- The main limitation I see is that of additional "pre-processing costs". As the authors point out in the limitations, if a new video is tested, their approach requires the addition of the video to the tree structure, which is quite costly. Whereas in existing approaches there is no such cost and more flexbility. Moreover, while in existing approaches the video encoding is a single step (extract features), here there are potentially three steps (extract CLIP features, add to the tree, extract features with the model used for retrieval/ranking). While they still achieve their goal (that is, improving online retrieval efficiency in a predefined gallery), these limitations are important in real scenarios where new videos may appear at any time.

- Few things which are not clear to me:

* At lines 387-390, the authors mention using the truncated version of the path as SemID. How is it truncated? Is it truncated at a fixed length, as in max depth of the tree? Or is it truncated with respect to some other information? Currently, this aspect is not clear. Moreover, when defining the truncated path L^t_path, t is used although it is not the same as query t, creating some confusion.

* The sentence at lines 470-473, "In the decoding process, considering the semantic summarization ability of natural language query, which makes videos that meet the description may not be in the same path, we use the Beam Search algorithm for decoding", makes little sense to me.

**Suitability:**

3

---

### Official Review · Reviewer_VtG3 · 2024-05-26

**Rating:** 4
**Confidence:** 3

**Summary:**

This paper introduces a novel approach named T2VIndexer which aims to improve the efficiency of existing text-to-video retrieval methods and maintain their performance simultaneously. It proposes a encoder-decoder framework which build a video semantic tree first and allocate the semantically similar videos to the same leaf. Then the decoder will predict the path to the leaf through the given query and obtain candidate videos and the number of it is far less than the origin candidates. To this end, the final retrieval process on the remain videos is remarkable faster than before. Besides, extensive experiments on four benchmarks shows the effectiveness on retrieval efficiency enhancement and performance retention of the method.

**Strengths:**

1.The motivation of this paper is well-founded, the idea of build video semantic tree to cluster similar videos and training a network to filter candidate videos by path searching is novel and convincing .
2.The experiments are sufficient and have demonstrated the effectiveness of the proposed method. It also greatly boosting the retrieval efficiency and outperforms previous works.
3. Moreover, the paper is well-written and presents its concepts clearly and comprehensibly which makes it easy to follow.

**Limitations:**

1. This paper tends to employ parameter-efficient learning strategies for text-video retrieval, however, the parameter-efficient learning methods in recent years are not mentioned in the related work.
2. Constructing video semantic tree by k-means is poor in interpretability, how about using concept-based methods?
3. Since the Vi-Sem Tree is built based on an individual dataset, I think generalization ability of it is weak.
4. In Sec 4.4, how to construct model without query expansion and  Without Vi-SemTree and SemID is not mentioned.

**Suitability:**

3

---

### Official Review · Reviewer_u8tp · 2024-05-26

**Rating:** 4
**Confidence:** 2

**Summary:**

This work studies text-based video retrieval tasks and particularly focuses on boosting cross-modal retrieval efficiency. Specifically, the authors propose a novel model-agnostic module based on a multimodal representation foundation model (CLIP), multimodal LLM, and hierarchical k-means algorithm to improve the inference performance and efficiency of VTR models. By building a tree (named Video Semantic Tree in the manuscript) with CLIP and clustering algorithm, the video gallery is turned into a set of small semantic-related groups and each group can be encoded as a specific index, thus effectively mitigating the burdens in dual stream VTR paradigms. The authors evaluate the proposed T2VIndexer at four benchmark datasets combined with different baseline models. Experimental results demonstrate the effectiveness of both efficiency and performance.

**Strengths:**

Strength:
1. The motivation is interesting and practical, which can be applied to real application scenarios.
2. The writing is generally clean and most ideas have been clearly presented. The figures are also informative.
3. Extensive experiments demonstrate the effectiveness of the proposed work.

**Limitations:**

Weakness:
1. The authors may want to present their code but it seems that the repo had been expired. No code was found there. Please consider providing the correct URL.

2. The authors are encouraged to add more discussions on generative retrieval models or efficiency-sensitive video content retrieval models, such as [1-4], that share similar ideas with this work, including building trees or turning online computation into an offline phase.

[1] Generative Retrieval with Semantic Tree-Structured Item Identifiers via Contrastive Learning. ArXiv'23

[2] Learning Commonsense-aware Moment-Text Alignment for Fast Video Temporal Grounding. ACM TOMM'24

[3] Faster Video Moment Retrieval with Point-Level Supervision. ACM MM'23

[4] CenterCLIP: Token Clustering for Efficient Text-Video Retrieval. SIGIR'22

**Suitability:**

3

---

### Official Review · Reviewer_D72x · 2024-05-30

**Rating:** 5
**Confidence:** 3

**Summary:**

The paper designs a Video Semantic tree to identify a video named Vi-SemTree to improve the efficiency of text-video retrieval independent of the number of candidate videos. It proposes a novel video indexer named T2VIndexer. For constructing Vi-SemTree and obtaining SemID sequence, the work extracts frame representation of the video by CLIP encoding, then integrates the representations into semantic-level information, and forms tree structures for identifying the videos and semID Encoding. The work introduces a sequence-to-sequence generative model to embed the input query into the transform encoder and output SemID sequences as tokens generated to generate the corresponding SemID based on the input query. Furthermore, the paper discusses the strategy of model training and inference. Experiments show that the retrieval time is sped, and the R@1 metric is improved. The idea is interesting and practical. Of course, the accuracy and robustness deserve further discussion.

**Strengths:**

(1) The paper is well-written and easy to follow;
(2) The paper introduces details around T2VIndexer construction, generation, inference, assignment, pre-select, and matching;
(3) The paper proposes a Vi-SemTree structure for Video Identification based on Video semantic representation and encoding. Video indexing does speed up text video retrieval efficiency in principle.
(4) The paper exploits Multi-modal Large Language Model(MLLM) tools to generate multi-view descriptions from rich semantic information in Video for multiple different.
(5) The paper introduces a Prefix-Aware Weight-Adaptor (PAWA) decoder to generate tokens at different positions for expressing different concepts due to their different prefixes and identify different representations at different positions.

**Limitations:**

(1)If the order of Vi-SemTree structure impacts indexing efficiency and text-video retrieval accuracy, please provide a detailed analysis and presentation.
(2)The paper mentions that both hierarchical Vi-SemTree construction and Vi-SemTree leaf node insertion depend on video semantic similarity. Please clarify the definition of semantic similarity metrics and their impact on accuracy and ranking results。
(3)In the context of the video semantic tree construction mechanism, if there are multiple videos in the leaf node corresponding to the same SemID, how should they be processed and matched? Could you provide some insights on this issue?

**Suitability:**

3

---

### Meta-Review · Area_Chair_d7RK · 2024-07-03

**Recommendation:** Accept (Oral)
**Confidence:** 5

**Metareview:**

This paper introduces a model-based video indexer named T2VIndexer, aiming to reduce retrieval time while maintaining high accuracy. It is a sequence-to-sequence generative model which can directly generate video identifiers and retrieve candidate videos with constant time complexity. It can enhance the retrieval efficiency of current state-of-the-art models on several datasets.

(+) On the positive side, the reviewers found the method to be interesting and effective and appreciate the clear motivation and good presentation .

(-) On the negative side, there are still some concerns on the pre-processing costs, insufficient discussion, generalization ability, and clarity of the writing.

Several technical questions and suggestions were raised by the reviewers. The authors have taken these into consideration. Overall, all reviewers agreed to accept the paper after the rebuttal. Therefore, the AC recommends accepting the paper.